# Biocontrol Effect of *Clonostachys rosea* on *Fusarium graminearum* Infection and Mycotoxin Detoxification in Oat (*Avena sativa*)

**DOI:** 10.3390/plants12030500

**Published:** 2023-01-21

**Authors:** Alfia Khairullina, Nikola Micic, Hans J. Lyngs Jørgensen, Nanna Bjarnholt, Leif Bülow, David B. Collinge, Birgit Jensen

**Affiliations:** 1Division of Pure and Applied Biochemistry, Lund University, 221 00 Lund, Sweden; 2Department of Plant and Environmental Sciences and Copenhagen Plant Science Centre, University of Copenhagen, DK-1871 Frederiksberg, Denmark

**Keywords:** oat, biocontrol, Fusarium head blight, deoxynivalenol, mycotoxins, *Clonostachys rosea*

## Abstract

Oat (*Avena sativa*) is susceptible to Fusarium head blight (FHB). The quality of oat grain is threatened by the accumulation of mycotoxins, particularly the trichothecene deoxynivalenol (DON), which also acts as a virulence factor for the main pathogen *Fusarium graminearum*. The plant can defend itself, e.g., by DON detoxification by UGT-glycosyltransferases (UTGs) and accumulation of PR-proteins, even though these mechanisms do not deliver effective levels of resistance. We studied the ability of the fungal biocontrol agent (BCA) *Clonostachys rosea* to reduce FHB and mycotoxin accumulation. Greenhouse trials showed that *C. rosea*-inoculation of oat spikelets at anthesis 3 days prior to *F. graminearum* inoculation reduced both the amount of *Fusarium* DNA (79%) and DON level (80%) in mature oat kernels substantially. DON applied to *C. rosea*-treated spikelets resulted in higher conversion of DON to DON-3-Glc than in mock treated plants. Moreover, there was a significant enhancement of expression of two oat UGT-glycosyltransferase genes in *C. rosea*-treated oat. In addition, *C. rosea* treatment activated expression of genes encoding four PR-proteins and a WRKY23-like transcription factor, suggesting that *C. rosea* may induce resistance in oat. Thus, *C. rosea* IK726 has strong potential to be used as a BCA against FHB in oat as it inhibits *F. graminearum* infection effectively, whilst detoxifying DON mycotoxin rapidly.

## 1. Introduction

Oat (*Avena sativa*) is an important food and fodder crop in Northern Europe and Northern America [1,2]. The majority of the oats produced is used for livestock feed, but human consumption of oats has increased during the past decade due to its high nutritional value, good taste, recognised health benefits and use in new food products such as oat milk [3,4,5]. Thus, oat is emerging as a valuable crop in the transition towards a plant-based diet. However, oats are vulnerable to the disease Fusarium head blight (FHB) caused by a complex of *Fusarium*-species including *F. graminearum, F. avenaceum, F. poae, F. langsethiae* and *F. culmorum*. Although these fungal pathogens can cause substantial yield losses, their main threat lies in their production of contaminating mycotoxins, which compromise food and feed safety [6,7,8,9].

During the last decade, FHB infection in oat has been high with *F. graminearum* as the dominating species and deoxynivalenol (DON) as the prevalent mycotoxin [7,10,11,12]. DON is a type B trichothecene, which inhibits protein biosynthesis [13,14] and can cause both acute and chronic toxicoses in humans and livestock. Another important mycotoxin produced by *F. graminearum* is zearalenone (ZEA), a mycoestrogen causing reproductive disorders in mammals [15]. To protect human and animal health, the European Food Safety Authority (EFSA) established maximum limits for DON and ZEA in cereals and cereal products. Thus, maximum EU levels for DON and ZEA in unprocessed oats are 1750 µg/kg and 100 µg/kg, respectively [16,17].

Fungicides sprayed at anthesis can reduce FHB and DON levels in cereals [18,19,20]. However, there are rising concerns about their use due to the increase of fungicide resistance and accumulation of fungicide residues in the environment. Moreover, as it takes over a week for an oat panicle and about a month for the whole plant to go through anthesis [21], finding optimal fungicide application times for the treatment of FHB in oat is a difficult task. Disease management using microbial biological control agents (BCAs) has gained momentum recently as it is considered to offer a sustainable and environmentally friendly path to crop production [22]. For the control of FHB in cereals, the fungus *Clonostachys rosea* has been tested in several trials. In Canadian field trials, isolate ACM941 of *C. rosea* showed a significant reduction of FHB in wheat as well as a 22–33% reduction in the DON content following spray treatments at flowering [23,24]. In addition, spraying an oil-based formulation of *C. rosea* isolate SHA77.3 reduced FHB in Swiss field trials in one of two tested wheat cultivars whereas the DON content was reduced in both cultivars by 45–69% [25]. However, despite problems with FHB and mycotoxin accumulation in oats, there are, to the best of our knowledge, no reports on the use of BCAs to combat the disease in oat production.

DON plays an important role for the pathogen, by acting as a virulence factor during *Fusarium*-infection [26,27]. However, plants can reduce the phytotoxic effect of DON by detoxification reactions, with conjugation into the much less acutely toxic DON-3-glucoside (DON-3-Glc) by UDP-glucosyltransferases (UTGs) as the main known mechanism [28,29,30]. Thus, the overexpression of the barley gene *HvUGT13248* in wheat reduced levels of DON (and nivalenol) and simultaneously decreased disease severity of both FHB and Fusarium crown rot [31,32]. DON-3-Glc is categorised as a ‘masked mycotoxin’, a term used to describe plant metabolites of mycotoxins, which can potentially be toxic after ingestion by mammals [33]. For example, DON-3-Glc was shown to be hydrolysed in rats by their gut microflora to release DON [34]. Cereals, including oat, can also detoxify DON by conjugating with other sugars, glutathione and other substances, but to a much lower extent [29,35].

Recently, we identified and characterised two oat DON-detoxifying UGTs, *AsUGT1* and *AsUGT2*, orthologous to barley *HvUGT13248* [36]. As increased DON detoxification has been linked directly to increased resistance to FHB in cereals, finding ways to enhance DON-detoxifying capacity in plants could contribute to developing resistance against FHB in the field. In addition to using cultivars with confirmed high DON-detoxifying ability, an equivalent DON-detoxifying response in a plant could, hypothetically, be induced by the application of BCAs.

It has been widely demonstrated that fungal BCAs can activate plant defence genes that play important roles in hampering pathogen infection, such as those encoding phytoalexin production and pathogenesis-related proteins (PR-proteins) [37]. The BCA *C. rosea* can induce expression of PR-genes encoding PR-proteins in wheat [38] and tomato [39]. These proteins are synthesised in plants in response to pathogen infection or exposure to abiotic stress. Several PR-proteins have antifungal properties and their induction is observed in plants exhibiting a high level of disease resistance. In common with most pathogens, *F. graminearum* is known to upregulate PR-genes in cereals [40,41,42], suggesting their role in plant defence against the pathogen.

WRKY transcription factors have likewise been reported to play a major role in plant disease resistance [43]. WRKY transcription factors were upregulated in wheat and barley tissues in response to *F. graminearum* infection [44,45] and specifically, WRKY23 and WRKY70 were shown to participate in activating defence against *F. graminearum,* correlating with resistance [46,47]. It was demonstrated that *C. rosea* upregulates genes in the WRKY family and other transcription factors in tomato [48,49,50], but, to our knowledge, no such transcription factor activation by *C. rosea* has been demonstrated in cereals. 

In the current study, we used the *C. rosea* strain IK726, isolated from barley roots and reported to act as an effective BCA to control several plant pathogens [51,52], including pathogenic *Fusarium*-species causing seedling blight in wheat and barley. Recently, it has also been demonstrated that *C. rosea* IK726 reduced FHB symptoms as well as DON content in wheat both in greenhouse tests and in the field [53,54]. We show that application of *C. rosea* IK726 to oat spikelets: (i) reduces both *F. graminearum* biomass and mycotoxin content in the spikelets, (ii) enhances expression of two oat UGT genes and DON-glucosylation in response to DON application and (iii) activates the expression of genes encoding four PR-proteins and a WRKY transcription factor.

## 2. Results

### 2.1. C. rosea Reduces F. graminearum Biomass and Mycotoxin Content in Mature Oat Kernels

In order to elucidate a potential biocontrol ability of *C. rosea* strain IK726 against FHB in oat, three independent greenhouse experiments were performed where oat spikelets were inoculated first with spores of *C. rosea* and subsequently with *F. graminearum* spores under conditions favouring *Fusarium*-infection. Quantifying *F. graminearum* DNA after harvest showed that *C. rosea* significantly reduced the amount of *F. graminearum* DNA in mature grain (Figure 1A). The amount of *F. graminearum* DNA in three trials was reduced in the range of 69–97%, with an average reduction of 79% in *C. rosea*-treated spikelets. Special care was taken to avoid contamination by droplets of fungal conidial suspension; no *F. graminearum* DNA was detected in control treatments (mock (no *F. graminearum*)/mock and *C. rosea*/mock, results not shown).

We quantified 3-ADON, DON-3-Glc and DON in the infected oat kernels using LC-MS/MS. This analysis showed that DON accumulation (sum of DON, 3ADON and DON-3-Glc) was significantly reduced by 62–93% in three trials in *C. rosea*-treated compared to mock-treated spikelets (Figure 1B and Table 1). Taken separately, DON was reduced by 64–93% (significantly in all three trials), 3ADON was reduced by 53–95% (significantly in two trials), and DON-3-Glc was reduced by 37–78% (significantly in one trial). Zearalenone (ZEA) was found in the samples in much lower quantities (10–30 pg/mg) compared to DON, but *C. rosea* treatment did not have a significant effect on the level of this mycotoxin (data not shown).

The calculated percentage of DON-3-Glc relative to the total level of DON showed a tendency of *C. rosea* treatment to increase the conversion of DON into DON-3-Glc, but a significant increase was observed only in trial 3 (Figure 1F).

### 2.2. C. rosea Enhances Conversion of DON into DON-3-Glc in Oat Spikelets

In the experiment with infected oats, it was difficult to assess the effect of *C. rosea* on glucosylation of DON, as the suppression of the infection led to variable amounts of DON in the samples. To obtain a more reliable comparison of levels of conversion of DON into DON-3-Glc in *C. rosea*-treated and mock-treated oat spikelets, DON was applied to oat spikelets and the compounds were quantified at 24, 48, 72 and 96 h after application. LC-MS/MS analysis showed that, in spikelets treated with *C. rosea*, DON was conjugated into DON-3-Glc at a much higher level by 24 h compared to non-treated spikelets (Figure 2) and remained high through the course of the experiment. In addition to DON-3-Glc, compounds with molecular masses corresponding to three more DON conjugates were detected: DON-diglucoside, DON-hexitol and 15-acetyl-DON-glucoside that are all described DON-detoxification products [29]. Due to lack of standards, these metabolites could not be quantified, but judging from the areas of the chromatogram peaks (Appendix A) higher amounts of DON-diglucoside and 15-acetyl-DON-glucoside were formed in *C. rosea*-treated spikelets at early time points compared to the water treated control. The peaks were small compared to those of DON and DON-3-Glc; as all compounds are glycosidic derivatives of DON, their ionization efficiencies in the LC-MS analysis can be assumed to be roughly similar, indicating that DON-3-Glc is the main DON detoxification product. DON-glutathione conjugates were previously identified in *Fusarium*-infected mature oat grain (unpublished data), but not detected here in the DON treated spikelets.

### 2.3. C. rosea Enhances Expression of Oat UDP-glucosyltransferases in DON-Treated Oat Spikelets

Samples from the same experiment as described above, where oat panicles were inoculated with *C. rosea* and treated with DON after 3 days, were used to study the effect of *C. rosea* on the expression of the two previously identified UGT genes [36] in the treated oat spikelets. Spikelets for qPCR analysis were collected at 0, 2, 4, 8, 12 and 24 h after DON treatment. Quantification of the UGT transcripts showed that the accumulation of both *AsUGT1* and *AsUGT2* transcripts at all time points was significantly higher in *C. rosea* than in mock-treated spikelets (Figure 3).

Interestingly, already at 2 h after DON application, the expression of *AsUGT1* was 6 fold higher and the expression of *AsUGT2* 7 fold higher in *C. rosea*-treated compared to mock-treated spikelets. Induction of UGTs in *C. rosea*-treated spikelets at 4 h after DON application was as high as the induction of UGTs at 12 h in water-treated spikelets. 

### 2.4. C. rosea Induces Expression of Genes of Oat PR-Proteins and WRKY23 Transcription Factor

To shed light on the mode of biocontrol activity of *C. rosea* against *F. graminearum* in oat, the expression of genes coding for four PR-proteins and two WRKY transcription factors were analysed in oat spikelets inoculated with *C. rosea* or left untreated (mock-treatment). *C. rosea* treatment significantly induced expression of the genes *PR1* (anti-fungal) by 58 fold, *PR3* (chitinase) by 30 fold, *PR4* (wheatwin) by 69 fold and *PR5* (thaumatin-like) by 27 fold compared to mock treatment (Figure 4). Expression of the *WRKY23-like* gene was significantly induced by 7 fold, while expression of the *WRKY70-like* gene did not change significantly. 

## 3. Discussion

While oat is affected by FHB to a similar degree as other cereals in terms of yield losses and mycotoxin accumulation, biological control of FHB in oat has been neglected. Previously, *C. rosea* has found to control *Fusarium* infection in several crops, including wheat [23,24,38] and maize [55,56]. Here, we studied the ability of *C. rosea* IK726 to control FHB and reduce mycotoxin content in oat and investigated the effect of *C. rosea* on DON glucosylation. Additionally, we examined the ability of *C. rosea* to induce the expression of certain PR proteins and WRKY transcription factors, indicating that mechanisms of control could involve induced resistance.

### 3.1. C. rosea Reduces F. graminearum Biomass and Mycotoxin Content in Mature Oat Kernels

Application of *C. rosea* IK726 to oat spikelets at anthesis at 3 days prior to *F. graminearum* inoculation reduced *Fusarium* biomass in three trials substantially (69–97% reduction). In wheat, where the biocontrol ability of *C. rosea* strain ACM941 on FHB has been studied previously, the effect was quantified as the percentage of *Fusarium* damaged kernels (FDK) [23,24]. In greenhouse trials, reductions of FDK at 65% and 68–92% were found, respectively. In wheat, there is a high correlation between FDK and *F. graminearum* biomass [25,57]. However, symptoms of FHB in oat are often quite cryptic due to the hulls covering the seeds, which makes a visual quantification of FDK unreliable. Therefore, in the present work, we quantified the level of *Fusarium* infection by amount of *Fusarium* DNA in oat spikelets. *C. rosea*-mediated biocontrol efficacy against FHB in oat is potentially at the same level as the efficacy obtained with *C. rosea* in wheat. 

Treatment with *C. rosea* also reduced levels of DON considerably (64–93%) in mature oat kernels. This is in line with DON reduction observed in wheat [23,24,25] and further strengthens the potential of *C. rosea* for biocontrol of FHB in cereals. *F. graminearum* strains initially synthesise an acetylated form of DON, i.e., 3-ADON or 15-ADON depending on the chemotype [58,59], which are later deacetylated into DON in the plant [60]. Strain LS G2 is a 3-ADON producer and DON-3-Glc was the main DON detoxification product identified when oat spikelets were inoculated with DON (Figure 3). Therefore, we consider that DON, 3-ADON and DON-3-Glc, roughly represent the bulk of all DON and DON derivatives produced by *F. graminearum* in the inoculated spikelets. Collectively, the sum of DON, 3-ADON and DON-3-Glc produced during the infection was reduced by 62–93% in *C. rosea*-inoculated compared to mock-treated spikelets. 

Whereas reduction of DON accumulation is the main goal of biocontrol of *Fusarium* spp. in cereals, the potential ability of *C. rosea* to affect the detoxification of DON is also relevant. The percentage of DON-3-Glc relative to the sum of DON, 3ADON and DON-3-Glc was higher in *C. rosea*-treated compared to mock treated spikelets, although differences were significant only in one of three trials (Figure 1E). The higher percentage of DON-3-Glc in the *C. rosea* treated spikelets is an indicator of enhanced DON-detoxification occurring in the plant cells. This could occur either by *C. rosea*-mediated upregulation of plant UGTs as demonstrated in this study, the involvement of *C. rosea* glucosylation enzymes or a combination of both. As for the ability of *C. rosea* to detoxify DON, recently, 15-acetyl-DON-3-Glc was reported in the interaction zone between *C. rosea* and *F. graminearum* in a dual culture assay [61]. Whether such mechanisms are active *in planta* remains to be determined. 

In a previous study, Gimeno et al. [25] found that *C. rosea* either did not have any effect on ZEA accumulation in wheat grain or in some cases increased it. In our three trials, ZEA was detected in low quantities and *C. rosea* did not have any effect on ZEA accumulation. However, ZEA production appears to be highly dependent on climate conditions, especially the amount of rainfall [62,63]. Since we performed our trials in the greenhouse, the conditions were most probably more favourable for DON than for ZEA production. 

### 3.2. Treatment of C. rosea-Inoculated Spikelets with DON Increased Conversion of DON into DON-3-Glc and Enhanced Expression of Two Glucosyltransferase Genes

To investigate the fate of DON and expression of DON-detoxifying genes in green oat spikelets, we applied a high concentration of DON to spikelets pre-treated with *C. rosea*. By applying only the mycotoxin DON rather than *F. graminearum* itself, the time needed for the pathogen to produce large amounts of DON (up to 144 h) [44] was eliminated. At the same time, the possibility of natural senescence in the oat spikelet tissues occurring during the infection period was excluded. 

Conjugation of DON into DON-3-Glc at all time points occurred to a much higher degree in *C. rosea*-treated spikelets compared to mock-treated spikelets. The largest difference was observed at the earliest time point, 24h post-DON application, when the level of DON-3-Glc was 14 times higher in *C. rosea*-treated spikelets than in mock-treated. 

Demissie et al. [61] found that *C. rosea* induces formation of 15-ADON-3-Glc in confrontation assays with *F. graminearum*, which suggests an upregulation of glucosyltransferases in *C. rosea*. In the current experiment, we cannot exclude the possibility that *C. rosea* enzymes contribute directly to DON-glycosylation in plant tissues, although this contribution is unlikely to be as pronounced as that of a plant’s own detoxifying machinery. *C. rosea* possesses a very effective mechanism for removing toxic substances from the cells into the apoplast in form of numerous drug membrane transporters [52,61,64]. Therefore, this fungus is probably less dependent on DON-detoxification mechanisms than plants.

High levels of DON-3-Glc are most probably the result of induction of UGT genes in plant tissues, as we observed a significantly enhanced expression of the two oat UTG genes (*AsUTG1* and *AsUTG2*) in *C. rosea*-treated samples. These two UGTs were previously characterised and found to be highly inducible by both DON-treatment and *F. graminearum* infection [36]. The presence of *C. rosea* led to significantly higher induction of the oat UGTs in DON-treated spikelets at all time points after DON-application. Effective and fast DON-detoxification, especially in earlier stages of infection, is an important component of resistance against *F. graminearum* infection [29]. Such a considerable and rapid response in *C. rosea*-treated tissues could better prevent protein biosynthesis damage caused by binding DON to the ribosomes. To the best of our knowledge, we demonstrated for the first time the activation of cereal UTGs by a fungal BCA. 

The two oat UGT genes were not directly induced by *C. rosea*. Thus, 3 days after *C. rosea* inoculation and immediately before DON application (Figure 3A,B, 0 h), expression of the genes was low and did not differ between *C. rosea* and mock treatments. Enhanced DON-induced expression of these genes in *C. rosea*-treated spikelets could be the result of *C. rosea* upregulating the expression of various transcription factors, preceding the expression of the defence genes. 

Furthermore, *C. rosea* might directly or indirectly upregulate genes of other plant DON-detoxifying enzymes. While no glutathione (GSH) conjugates were detected, we detected compounds with molecular masses corresponding to DON-diglucoside, 15-acetyl-DON-glucoside and DON-hexitol, which were previously found in DON-treated wheat spikes among several other conjugates [29]. These compounds could not be quantified, but the two first are glucosides and likely derivatives of DON-3-Glc, underscoring the importance of the UGT-mediated pathway for DON detoxification in oat.

### 3.3. C. rosea Inoculation Induces Expression of PR-Genes and a WRKY Transcription Factor

Induction of PR-genes by BCAs as a first line of plant resistance is well known [65], but has not been studied previously in oat. In a hexaploid species like oat, there are numerous gene copies within each PR-protein family, which could be expressed in addition to those that we have quantified. Transcripts of genes corresponding to the four oat PR-proteins accumulated to high levels (27–69 fold increase) in spikelets treated with *C. rosea* at 3 days prior to sampling as compared to the expression of these genes in spikelets treated with water only. In the biocontrol trials (Figure 1), *F. graminearum* inoculum was applied at 3 days after inoculation with *C. rosea* and therefore we assume that genes coding for these PR-proteins were already expressed at that time point. Previously, transcripts for PR1, PR4 and PR5 were found to accumulate in coleoptiles and roots of wheat when these tissues were inoculated with *C. rosea* [38]. Homologues of the four PR-proteins studied here in response to *C. rosea* treatment are known to play important roles during *F. graminearum* infection in other cereals. Thus, genes belonging to the PR1-protein family were specifically induced in *F. graminearum*-inoculated maize [66]; transgenic wheat, expressing chitinase II (PR3), exhibited increased resistance against *F. graminearum* [67]; PR4-protein was shown to inhibit *F. graminearum* development in wheat kernels [68] and PR5-protein was expressed in a resistant barley cultivar after *F. graminearum* inoculation [69]. Substantial accumulation of these four PR-protein transcripts in oat spikelets after treatment with *C. rosea* suggests that protection against *F. graminearum* in spikes may at least partly include induced resistance. However, whether *C. rosea* can induce resistance against *F. graminearum* would need to be tested. This would be the case if rapid and enhanced expression of defence-related processes occur after pathogen arrival [65]. To the best of our knowledge, such a strong response in PR-genes, mediated by the presence of *C. rosea* alone, has not been demonstrated for BCAs previously.

WRKY transcription factors are known to play an important role in plant defence responses as they regulate expression of various defence genes [43]. WRKY70 was shown to have a positive effect on the resistance against *F. graminearum* in wheat [46]. Recently, WRKY23 was found to be involved in modulating defence responses against FHB in barley by regulating genes important for reinforcing cell walls against the invasion of pathogen, including a glucosyltransferase gene [47]. To determine the ability of *C. rosea* to upregulate WRKY transcription factors, relevant in defence against *F. graminearum*, we examined expression of the *WRKY23-like* and *WRKY70-like* genes in oat spikelets treated with *C. rosea*. While we did not observe any significant increase in transcript accumulation in *C. rosea*-treated oat spikelets for the *WRKY70-like* gene, WRKY23-*like* transcript accumulation was significantly increased by 7 fold. This suggests that one of the modes of biocontrol action of *C. rosea* could be an upregulation of transcription factors involved in the expression of defence genes.

## 4. Conclusions

We showed that treatment with *C. rosea* IK726 substantially reduced both *F. graminearum* biomass and mycotoxin content in oat kernels. *C. rosea* appears to upregulate different types of defence in plant tissues. While defence genes, such as those encoding four PR-proteins and WRKY23-like transcription factors were activated directly, induction of UGT encoding genes was indirect and could occur due to *C. rosea*-mediated pre-activation of certain transcription factors. *C. rosea*-mediated induced resistance in oat spikelets helps the plant to not only effectively inhibit the *F. graminearum* infection, but also to detoxify mycotoxins produced by the pathogen more rapidly, demonstrating the versatile potential of treatment with this BCA. Hence, this strain has a strong potential to be used as a BCA against FHB in oat. Future experiments should investigate the durability of *C. rosea* for reduction of FHB and mycotoxin content in oat under field conditions.

## 5. Materials and Methods

### 5.1. Plant and Fungal Material

Oat cv Belinda was cultivated as follows: two seeds per pot (16 cm in diameter) were planted in a mix containing soil (*Krukväxtjord med lera och kisel*, SW Horto, Sweden), 9% (*v*/*v*) perlite and 0.3% (*v*/*v*) Basacote plus 3M granulated fertilizer (N-P-K[+Mg+S] 16–8-12 [+2+5], Compo Expert, Münster, Germany). Plants were grown in a greenhouse with the light intensity set at 300 µE m^−2^ s^−1^ at the panicle level and relative humidity 65%. The temperature was maintained at 22 °C during the 16 h day and 18 °C during the 8 h night. 

*C. rosea* strain IK726 [51] was cultured on potato dextrose agar (PDA, Scharlau Microbiology, Barcelona, Spain) plates for 7 days. Spores were harvested by flooding the plates with sterile MilliQ water with 0.02% (*v*/*v*) Tween 20 added. The suspension was filtered through a 100-µm cell strainer (Sarstedt, Nümbrecht, Germany) and the concentration of the spores was adjusted to 10^7^ spores/mL. 

*F. graminearum* strain LS_G2 (3-ADON chemotype) was previously isolated from a farm sample of *Fusarium*-damaged oat seeds grown in Sweden. The attribution of this strain to *F. graminearum* species was determined by sequencing of *EF-1α* and *RPB2* genes fragments according to O’Donnel et al., 2010 [70] and the chemotype was determined by PCR according to Quarta et al., 2006 [71]. Inoculum was obtained by culturing the fungus on PDA plates for 5 days and subsequently growth in mung bean broth [72] for another 5 days at 25 °C with shaking at 150 rpm. Macroconidia were filtered through a 100-µm cell strainer (Sarstedt, Nümbrecht, Germany), harvested by centrifugation at 3800 RCF for 10 min and washed once with sterile distilled water. Spores were resuspended in sterile water with added Tween 20 (0.02%, *v*/*v*). The concentration of macroconidia was adjusted to 10^6^ spores/mL.

### 5.2. Application of C. rosea, F. graminearum and DON to Flowering Spikelets

To evaluate the biocontrol effect of *C. rosea* and the corresponding effect on DON accumulation in mature grains, plants were inoculated at anthesis (GS 65), either with a spore suspension or with water containing 0.02% (*v*/*v*) Tween 20 (mock). Twenty-µL *C. rosea* spore suspension (10^6^ spores/mL) or water was pipetted in the space between two adjacent florets in a spikelet. Inoculated spikelets were marked with a waterproof marker. Panicles were covered with 3-L plastic bags, misted with distilled water beforehand. After 48 h, the bags were removed and panicles were sprayed with distilled water three times per day. At 72 h after inoculation with *C. rosea*, spikelets were inoculated with 15 µL *F. graminearum* spore suspension (10^5^ spores/mL) or with water containing 0.02% (*v*/*v*) Tween 20 (mock), in the same way as above. Treated oat panicles were handled in the same way as above, except that bags were removed after 72 h. A total of 15 spikelets were inoculated per panicle with 4 panicles per treatment. Spikelets were harvested at grain maturity (100 days after sowing). Thus, 15 spikelets from each panicle were pooled and represented one replicate. Thus, there were four replicates per treatment. Three separate greenhouse trials were performed over a six-month period of the same year. 

To study the DON-induced expression of oat UDP genes in *C. rosea*-treated spikelets, plants were inoculated with *C. rosea* as described above. At 72 h after inoculation, spikelets were treated with an aqueous DON solution (0.6 mg/mL) or with distilled water (mock). At time point zero, 10 µL DON solution was pipetted into the space between two adjacent florets in a spikelet. Treated spikelets were marked with a waterproof marker. Inoculated panicles were covered with 3-L plastic bags, sprayed with distilled water beforehand. A total of 10 spikelets were treated per panicle. At 0, 2, 4, 8, 12, 24 and 48 h after treatment, spikelets were sampled and immediately frozen in liquid nitrogen.

### 5.3. Quantification of F. graminearum DNA 

Marked spikelets from each panicle were harvested and hand-threshed (glumes were removed) and milled into a fine flour using a “Pulverisette 23” ball mill (Fritsch, Idar-Oberstein, Germany) set at 50 1/s oscillation for 4 min. DNA was extracted using the Bead Beat micro AX Gravity kit for genomic DNA purification (A&A Biotechnology, Gdansk, Poland). The quality and quantity of the extracted DNA were analysed spectrophotometrically and with Qubit Fluorometric Quantification (Invitrogen, Carlsbad, CA, USA) according to the manufacturer’s instructions. Primers and the probe 5′ 6-FAM/ZEN/3′IMFG (Integrated DNA Technologies, Leuven, Belgium) were designed to amplify a fragment of the *F. graminearum* TRI5 gene, based on the fragment from Waalwijk et al. [73]. A pair of primers with the probe 5′HEX/ZEN/3′IBFQ (Integrated DNA Technologies, Leuven, Belgium) were designed to amplify a fragment of the oat α-tubulin gene, which was used as a plant DNA reference gene (for primer sequences, see Appendix A). Duplex qPCR reactions targeting *F. graminearum* and the oat α-tubulin gene were performed in a final volume of 15 µL, containing 40 ng DNA, 750 pmol of each primer, 225 pmol of each probe and 7.5 µL PrimeTime™ Gene Expression Master Mix (Integrated DNA Technologies, Leuven, Belgium). Reactions were carried out with technical duplicates using a CFX96 Real-Time qPCR system (Bio-Rad Laboratories, Hercules, CA, USA). The thermocycler programme was set for 3 min at 95 °C followed by 42 cycles of 15 s at 95 °C, 20 s at 62 °C and 20 s at 72 °C. Default parameters of the Bio-Rad system were used to determine Ct-values for each reaction. Standard curves for DNA quantification were made using six 10-fold dilutions starting from 10 ng pure *F. graminearum* DNA and 50 ng pure oat DNA. For *F. graminearum* primers and probe, the sensitivity and efficiency of the assay were 1.5 pg and 97.3% and for oat α-tubulin primers and probes 5 pg and 102.3%, respectively. The ratio between the total amount of *F. graminearum* DNA and the total amount of oat DNA was calculated as the DNA load of *F. graminearum* in the samples. 

### 5.4. Analysis of DON-Induced UGT Expression in Oat

Experiments were performed in three biological repetitions. In each, RNA was extracted from two spikelets. Separate spikelets were ground under frozen conditions in pre-cooled (liquid nitrogen) plastic tubes with screwcaps (Sarstedt, Nümbrecht, Germany) and two 5 mm stainless steel beads (Qiagen, Hilden, Germany) using a Precellys Evolution homogenizer (Bertin Technologies, Montigny-le-Bretonneux, France) as follows: 25 s homogenisation at 5500 rpm, cooling in liquid nitrogen for 10 s and repeating the homogenisation for 25 s at 5500 rpm. RNA was extracted from the pool of two homogenised spikelets using the RNeasy Plant Mini kit (Qiagen, Hilden, Germany). The quality and quantity of the extracted RNA were analysed by agarose electrophoresis and spectrophotometrically. DNA was removed from the samples with DNA free Kit (Invitrogen, Waltham, MA, USA) and cDNA synthesis was performed using the iScript cDNA synthesis Kit (Bio-Rad Laboratories, Hercules, CA, USA). cDNA samples were diluted 1:20 (*v*/*v*) with TE buffer (10 mM Tris-HCl, pH 8.0; 1 mM EDTA), aliquoted and stored at −20 °C. Primers and the probe 5′ 6-FAM/ZEN/3′IMFG (Integrated DNA Technologies, Leuven, Belgium) were designed to amplify fragments of the *AsUGT1* and *AsUGT2* oat genes (for primer sequences, see Appendix A). The oat α-tubulin gene was used as a reference target and a pair of primers together with the probe 5′HEX/ZEN/3′IBFQ (Integrated DNA Technologies, Leuven, Belgium), amplifying a fragment of this gene, were used. The efficiency and sensitivity of the primers and probes in duplex reactions were previously tested [36]. The efficiency of the primers and probes was in range of 97–103% and the sensitivity was 0.1 ng of total oat RNA. Duplex qPCR reactions targeting one of the UGTs and the reference gene were performed in a final volume of 15 µL, containing 2 µL of the diluted cDNA product, 750 pmol of each primer, 225 pmol of each probe and 7.5 µL PrimeTime™ Gene Expression Master Mix (Integrated DNA Technologies, Leuven, Belgium). Reactions were carried out in technical duplicates using a CFX96 Real-Time qPCR system (Bio-Rad Laboratories, Hercules, CA, USA). The following thermocycler programme was used: 3 min at 95 °C followed by 42 cycles of 15 sec at 95 °C, 20 sec at 62 °C and 30 sec at 72 °C. The Ct-values were automatically determined for each reaction by the Bio-Rad system set with default parameters. The comparative ΔΔCt method was used to evaluate the relative quantities of each amplified product in the samples [74]. 

### 5.5. Analysis of the Expression of Oat PR-Proteins Genes and WRKY-Transcription Factors

Oat PR-protein and WRKY-transcription factor sequences were selected based on the sequence homology with previously characterised barley proteins: PR1 (X74939.1), PR3 (X78672.1), PR4 (Y10814.1), PR5 (AJ001268.1), WRKY23 (KT962219.1) and WRKY70 (MLOC_66134). We used the Sequenceserver v2.0.0 [75] tool to perform BLASTP [76] searches (E-value < 1.0 × 10^−5^) of the barley proteins against the oat cv. Sang v1.1 proteins. Additionally, we checked that the oat proteins and their corresponding barley homologues belong to the same orthogroups, previously reported by Kamal et al. [77].

Primers and the probe 5′ 6-FAM/ZEN/3′IMFG (Integrated DNA Technologies, Leuven, Belgium) were designed using the PrimerQuest Tool (Integrated DNA Technologies, Leuven, Belgium) to amplify gene fragments, corresponding to 4 oat PR-proteins and 2 WRKY transcription factors. Each primer pair and the probe were specific to two PR-gene copies and three WRKY gene copies in the hexaploid oat Sang genome. For primer sequences and the corresponding oat genes, see Appendix A. The oat α-tubulin gene was used as a reference target. qPCR reactions were performed as described above.

### 5.6. UPLC-MS/MS Analysis of DON, 3-ADON and DON-3-Glc in Mature Oat Kernels

The LC-MS/MS analyses were performed by the facility of the Swedish Metabolomics Centre, Umeå, Sweden. The following solvents were used: acetonitrile (ACN) and 2-propanol (IPA) hypergrade for LC-MS LiChrosolv (Merck, Darmstadt, Germany), formic acid, HiPerSolv Chromanorm for LC-MS (VWR Chemicals), and MilliQ gradient system purified H_2_O.

Mycotoxins were extracted by adding 1 mL solution containing 80% ACN (*v*/*v*), 0.1 ng/µL caffeine-13C3 (as an internal standard) to approximately 50 mg milled flour. The samples were shaken with one tungsten bead each at 30 Hz for 3 min in a mixer mill (MM 400, Retsch) and then centrifuged at 18,620× *g*) for 10 min at +4 °C. Fifty-µL supernatant was transferred to vials, dried under nitrogen gas flow and stored at −20 °C until analysis. Small aliquots of the remaining supernatants were pooled and used to make quality control (QC) samples. Blank samples, i.e., samples without starting material, were prepared the same way as the samples. 

The analyses were conducted using an Agilent UHPLC system (Infinity 1290) coupled with an electrospray ionization source (ESI) to an Agilent 6495 triple quadrupole system equipped with iFunnel Technology (Agilent Technologies, Santa Clara, CA, USA). Chromatographic separation was performed on a Waters UPLC HSS T3 column (2.1 mm × 50 mm, 1.8-μm particle size). The mobile phase consisted of 0.1% (*v*/*v*) formic acid in MQ-water (A) and 0.1% (*v*/*v*) formic acid in ACN:IPA (75:25) (B). 

Gradient: the flow rate was set to 500 μL/min and the column was heated to 40 °C. 100% A was run the first min followed by a linear increase of B to 70% over 2 min. A linear increase to 85% B under 1.5 min was followed by isocratic 85% B for 1.5 min. B was linearly increased to 99% for 0.5 min and held isocratic 1.5 min. The column was returned to its initial conditions in 0.3 min and re-equilibrated at 800 µL/min flow, for 1.5 min. Analysis parameters: sample injection volumes were 2 µL. The mass spectrometer was operated in both negative and positive ESI mode with gas temperature set at 150 °C; gas flow 16 L/min; nebulizer pressure 35 psi; sheath gas temperature 350 °C; sheath gas flow 11 L/min; capillary voltage 3000 V (neg) and 4000 V (pos); nozzle voltage 1500 V (neg) and 300 V (pos); iFunnel high pressure RF 90 V (neg) and 200 V (pos); iFunnel low pressure RF 60 V (neg) and 100 V (pos). The fragmentor voltage 380 V and cell acceleration voltage 5 V. For a list of dMRM transitions, see Appendix A. Data was processed using MassHunter Qualitative Analysis and Quantitative Analysis (QqQ; Agilent Technologies, Atlanta, GA, USA) and Excel software (Microsoft, Redmond, WA, USA). The following standards were used: the internal standard caffeine-13C3 (Cambridge Isotope Laboratories, Inc., Tewksbury, MA, USA) and the mycotoxin specific internal standards deoxynivalenol-13C15 (DON-13C15), 3-acetyl-deoxynovalenol-13C17 (3-AcDON-13C17), zearalenone-13C18 (ZEA-13C18), all purchased from Romer Labs, Austria. Internal standards were introduced to the samples at a final concentration of 0.5 ng/µL (caffeine-13C3 and ZEA-13C18) and 0.1 ng/µL, (DON-13C15 and 3-AcDON-13C17). A 13-level calibration curve of pure standards, comprising deoxynivalenol-3-glucoside, zearalenone and deoxynivalenol all purchased from Sigma/Merck, was prepared by serial dilution (range: 2.5 fg/µL–1500 pg/µL) and spiked with the internal standards (0.1 or 0.5 ng/µL).

### 5.7. UHPLC-MS Analysis of DON and its Conjugates in Green Oat Spikelets

Metabolites were extracted as follows: from each time point (of individual treatments), one spikelet was extracted in 200 µL 85% MeOH + 0,5% HCOOH (*v*/*v*), in 1.5 mL Eppendorf tubes, containing two steel balls and fitted in a tissue-lyzer for 2 min at 30 shakes/sec. The homogenates were centrifuged at 20,000× *g* for 10 min at 4 °C. The supernatant (methanol extract) was collected and stored at −20 °C until LC-MS analysis. 

For UHPLC-MS analyses, methanol extracts were diluted 2.5x with MilliQ + 0.5% (*v*/*v*) HCOOH. All samples were filtered using 0.22 µM PVDF multiwall filters at 3000 rpm for 5 min at 20 °C. Linamarin (Sigma Aldrich) was added as an internal standard to a final concentration of 20 µM.

UHPLC-MS analyses were performed using a Dionex Ultimate 3000RS UHPLC system (Thermo Fisher Scientific) coupled to a CompactTM (QqToF) mass spectrometer (Bruker Daltonics) equipped with a temperature controlled auto-sampler (10 °C) and column oven (set to 40 °C). Five-µL aliquots were injected onto a Kinetex XB-C18 UHPLC column (150 × 2.1 mm, 1.7 µm, 100 Å pore size; Phenomenex) and eluted with a flow rate of 0.3 mL/min. The mobile phase comprised solvent A (0.05% (*v*/*v*) HCOOH in water) and solvent B (0.05% (*v*/*v*) HCOOH in acetonitrile), eluted at an initial composition of 98% A and 2% B for 1 min, before a linear increase to 70% solvent B over 6 min and subsequent increase to 100% B over 2 min. The column was washed with 100% solvent B for 2 min, returned to 2% B over 1 min and subsequently re-equilibrated at 2% B for 4 min. The QqToF mass spectrometer was operated in electrospray full scan negative ion mode with the following instrument settings: m/z 50–1200: nebulizer gas (nitrogen), 2.5 bar; drying gas (nitrogen), 8 L/min; drying gas temperature, 220 °C; capillary voltage, 4500 V; spectra acquisition rate, 5 Hz. Data dependent MS/MS acquisition (collision energy 25 eV) was triggered for the 3 most intense ions in the full scan MS spectra. All LC-MS data were analysed using Compass DataAnalysis 4.3 software (Bruker Daltonik GmbH, Bremen, Germany).

### 5.8. Statistics Analyses

Data for *F. graminearum* biomass, mycotoxin accumulation, gene expression and LC-MSMS analyses were analysed by a mixed effect-model analysis of variance, with treatment as fixed effect and replication as random effect, assuming a normal distribution. If necessary, variances were stabilised by appropriate transformation. Hypotheses were rejected at *P* < 0.05. All analyses were performed in PC-SAS (release 9.4, SAS Institute, Cary, NC, USA).

## Figures and Tables

**Figure 1 plants-12-00500-f001:**
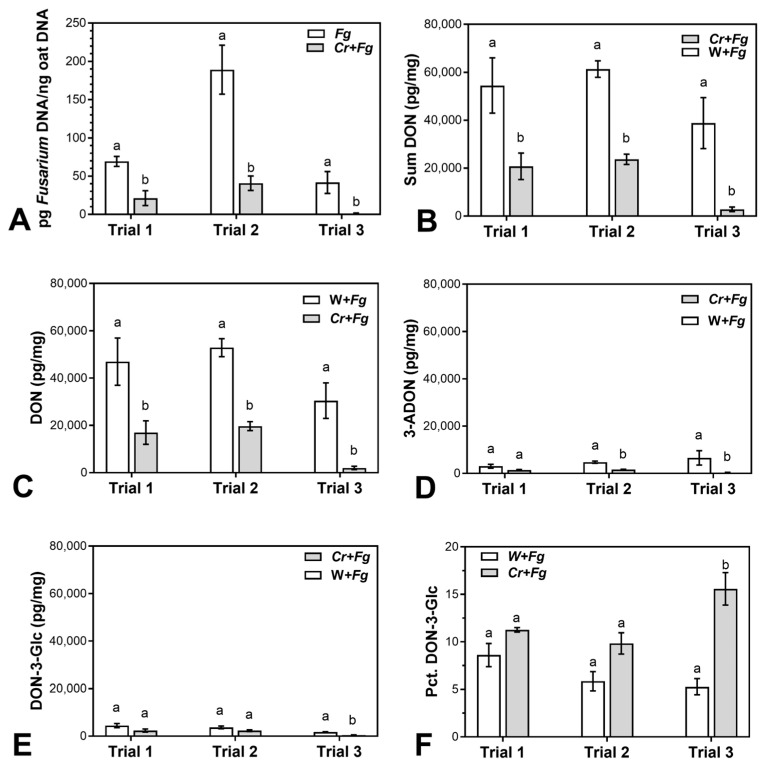
*Fusarium graminearum* biomass and mycotoxin levels in three independent experiments. (**A**) *F. graminearum* DNA amount (pg *Fusarium* DNA/ng oat DNA) in three independent trials. (**B**–**F**): Levels of three mycotoxins or masked mycotoxins in three independent trials: (**B**) sum of DON, 3-ADON, and DON-3-Glc, (**C**) DON, (**D**) 3-ADON, (**E**) DON-3-Glc, (**F**) Percentage of DON-3-Glc relative to the sum of DON, 3ADON and DON-3-Glc. Error bars represent standard error of the mean. Within each part of the figure, means marked with different letters are significantly different (*p* < 0.05).

**Figure 2 plants-12-00500-f002:**
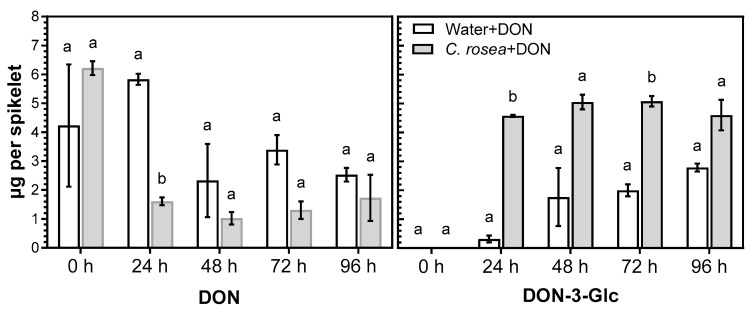
DON and DON-3-Glc in *C. rosea*-treated and mock treated oat spikelets. Spikelets were treated either with *C. rosea* followed by DON (*C. rosea* + DON), water+DON, *C. rosea*+water or water + water. DON and DON-3-Glc were only detected in *C. rosea*+DON and water + DON treated samples and the quantification of these two compounds is shown in the figure. Error bars represent standard error of the mean. Within each part of the figure, means marked with different letters are significantly different (*p* < 0.05).

**Figure 3 plants-12-00500-f003:**
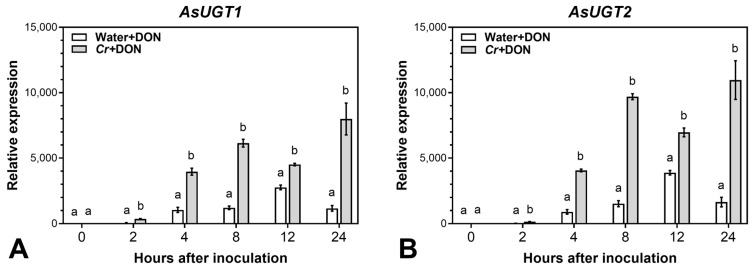
Relative expression of DON-induced (**A**) *AsUGT1* and (**B**) *AsUGT2* transcripts in *C. rosea*-treated and mock-treated oat spikelets. Error bars represent standard error of the mean. Bars marked with same letter are not significantly different (*p* ≤ 0.05).

**Figure 4 plants-12-00500-f004:**
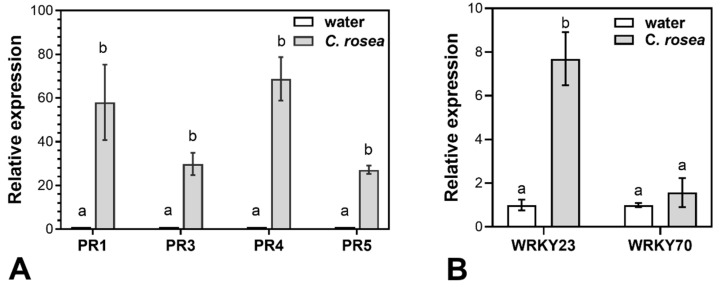
Expression of genes of four PR-proteins (**A**) and two WRKY transcription factors (**B**) in *C. rosea*-treated and water-treated oat spikelets 3 days after the inoculation. Error bars represent standard error of the mean. Bars marked with same letter are not significantly different (*p* ≤ 0.05).

**Table 1 plants-12-00500-t001:** Percent decrease of mycotoxins and amount of *F. graminearum* DNA (pg *Fusarium* DNA/ng oat DNA) in *C. rosea*-treated compared to non-treated spikelets and percent increase of DON3G relative to total level of DON.

Trial	*F. graminearum* DNA	Sum of DONs	DON	3-ADON	DON-3-Glc	DON-3-Glc/Sum of DONs ^1^
1	69.3	61.9	64.0	64.0	40.5	30.9
2	78.4	61.4	62.7	62.7	37.3	68.0
3	97.1	92.8	93.3	93.3	77.7	195.0

^1^ Percent DON3G relative to total level of DON (DON+3ADON+DON-3-Glc).

## Data Availability

Oat sequences, gene annotations, and orthogroups are available at https://doi.org/10.5447/ipk/2022/2 (accessed on 13 December 2022).

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
