# Peer review of "Biocontrol Effect of Clonostachys rosea on Fusarium graminearum Infection and Mycotoxin Detoxification in Oat (Avena sativa)"

_plants, 2023, doi:10.3390/plants12030500_

Round 1
Reviewer 1 Report
The manuscript is well written and generally clear. It brings interesting new data on how Fusarium head blight and mycotoxin (DON) prevalence in oats might be reduced by application by the biocontrol agent Clonotachys rosea. I believe it is a weakness that the study does not clearly demonstrate that the application of Fusarium in the presented experiments actually caused any pathogenesis-related symptoms on the plants. Only inhibitory effects of C. rosea towards Fusarium and its mycotoxin are shown. My major concern was with respect to the following. According to Materials and Methods (item 5.2), the experiments in Fig. 1 and Table 1 were performed on oat spikelets with or without application of C. rosea, followed by inoculation with or without Fusarium, so the way I understand it, there were four treatments: mock/mock, C. rosea/mock, mock/Fusarium and C. rosea/Fusarium. Nevertheless, there are only two treatments in each of the tables in Fig. 1. Why are the other treatments not shown? This information is very relevant, for example for the results in Fig. 1A. Was Fusarium DNA absent, for example, in the treatment(s) where no Fusarium was applied? Please, complement these result or explain why these data were omitted.
L. 20 and 125: I think the use of the term 'biomass' is misleading, because only fungal:plant DNA ratio was quantified and biomass was not directly assessed. Please make necessary adjustments.
Minor issues:
L. 470: Experiments were performed
L. 475: Improve description, not completely clear: '25 s homogenisation at 5500 rpm, cooling in liquid nitrogen followed by homogenisation.'
L. 476: 'pool of two pooled homogenised'??
L. 483: Were the primers tested (with respect to specificity for example)?
Author Response
The manuscript is well written and generally clear. It brings interesting new data on how Fusarium head blight and mycotoxin (DON) prevalence in oats might be reduced by application by the biocontrol agent Clonotachys rosea.
<< Thank you very much for finding time to review our manuscript, for the overall positive feedback and very good suggestions for clarifications and improvement of the text.
I believe it is a weakness that the study does not clearly demonstrate that the application of Fusarium in the presented experiments actually caused any pathogenesis-related symptoms on the plants. Only inhibitory effects of C. rosea towards Fusarium and its mycotoxin are shown.
<< If we understand this comment correctly, indeed we did not make a visual quantification of kernel symptoms of Fusarium disease. The symptoms of F. graminearum infection in oat are cryptic, not easily seen on living material, unless seeds are de-hulled. While in wheat or barley F. graminearum infection symptoms are easily accessible and quantifiable, in oat the symptoms are too variable, and it would be difficult to accurately access the effect of C.rosea judging by the symptoms. This is the reason why we quantify the Fusarium DNA as a measure of pathogen infection levels since we find that visual symptoms are unreliable in this pathosystem. As an illustration we have added Figure 2 in supplementary materials, showing an infected oat panicle and kernels both with hulls and de-hulled.
My major concern was with respect to the following. According to Materials and Methods (item 5.2), the experiments in Fig. 1 and Table 1 were performed on oat spikelets with or without application of C. rosea, followed by inoculation with or without Fusarium, so the way I understand it, there were four treatments: mock/mock, C. rosea/mock, mock/Fusarium and C. rosea/Fusarium. Nevertheless, there are only two treatments in each of the tables in Fig. 1. Why are the other treatments not shown? This information is very relevant, for example for the results in Fig. 1A. Was Fusarium DNA absent, for example, in the treatment(s) where no Fusarium was applied? Please, complement these result or explain why these data were omitted.
<< we agree that a full explanation is missing in our manuscript. We did not detect F. graminearum DNA in mock/mock and C.rosea/mock samples using a sensitive probe-based qPCR assay (detection limit 0.5 pg Fusarium DNA). Therefore, this data is not presented. The efficiency and specificity of primers were tested in our previous work, and this information is now added.
- 20 and 125: I think the use of the term 'biomass' is misleading, because only fungal:plant DNA ratio was quantified and biomass was not directly assessed. Please make necessary adjustments.
<< adjusted as suggested
The remaining suggestions are incorporated in the revised manuscript using track-changes:
Minor issues:
- 470: Experiments were performed
- 475: Improve description, not completely clear: '25 s homogenisation at 5500 rpm, cooling in liquid nitrogen followed by homogenisation.'
- 476: 'pool of two pooled homogenised'??
- 483: Were the primers tested (with respect to specificity for example)?
<< Primers and probes were tested and found to be specific and sensitive in our previous study [Khairullina, A.; Renhuldt, N. T.; Wiesenberger, G.; Bentzer, J.; Collinge, D. B.; Adam, G.; Bülow, L. Identification and functional characterisation of two oat UDP-glucosyltransferases involved in deoxynivalenol detoxification. Toxins 2022, 14,446]. https://doi.org/10.3390/toxins14070446
Kind regards,
The authors
Reviewer 2 Report
Abstract: Abstract is well written and concise. It contains all the necessary information such as aim, methods and the most important findings.
Line 22 and further in the text: perhaps control instead of mock would be better?
Introduction: Introduction is clear and smoothly leads to the aim of the study, which is distinctly presented.
Results: Results are presented clearly and according to scientific methods. The figures and tables are easy to understand and well described.
Table 1, lines 153-158 Maybe letters instead of *? Could you add SD to the table?
Discussion: The section is well written, and the findings are backed-up by literature.
Materials and Methods: They are clear, easy to follow and precise. The experimental design is sound. The results are reproducible.
Lines 404, 411: Please add PDA manufacturer.
The conclusions are consistent with the results.
Perhaps it would be beneficial if the text was checked by a native speaker or a professional service.
Author Response
Abstract: Abstract is well written and concise. It contains all the necessary information such as aim, methods and the most important findings.
<< thank you for the kind comments
Line 22 and further in the text: perhaps control instead of mock would be better?
<< We prefer the term „mock” as this is a more precise term than control. Mock means that the plants were treated in the same way as the inoculated plants, just without fungal inoculum.
Introduction: Introduction is clear and smoothly leads to the aim of the study, which is distinctly presented.
Results: Results are presented clearly and according to scientific methods. The figures and tables are easy to understand and well described.
<< thank you for the kind comments
Table 1, lines 153-158 Maybe letters instead of *? Could you add SD to the table?
<< The * were removed from this table since it gives a percentage effects for the various trials as they are not relevant for percentages
Discussion: The section is well written, and the findings are backed-up by literature.
Materials and Methods: They are clear, easy to follow and precise. The experimental design is sound. The results are reproducible.
The conclusions are consistent with the results.
<< thank you
Lines 404, 411: Please add PDA manufacturer.
<< added
Perhaps it would be beneficial if the text was checked by a native speaker or a professional service.
<< Among the co-authors of this manuscript, we happen to have a native speaker with four decades of experience of writing scientific articles and editing books. While revising this manuscript we made sure that this invaluable resource has been used to its fullest – a number of revisions have been made.
Kind regards,
the authors.
Reviewer 3 Report
COMMENTS TO THE AUTHOR REVIEWER
In this study, the authors investigated underlying biocontrol effect of biocontrol agent Clonostachys rosea against Fusarium graminearum infection and mycotoxin detoxification in oat by determining Fusarium biomass and DON level. It is meaningful to develop a biocontrol agent which can suppress mycotoxin accumulation of F. graminearum in oat. I think some of authors’ claims are properly supported by data of this manuscript. Thus, a modification of the language and interpretation of the observation may clarify the true nature of this work for the reader. Major comments: Try to synthetize abstract in a way it is more clear and concise.In general, improvement of the writing would also help to enhance this work. For example, the description of Clonostachys rosea in the introduction was extremely limited but would have been very helpful to understand the context of the work and why experiments were chosen.
The authors clarified that no reports on the use BCAs to combat the disease in oat production (line 66). The sentence lack of rigor is questionable.
In addition, I have reservations about the interpretation or replication of some of the experiments. It would also help to indicate within each figure legend how many experimental replicates and biological replicates were conducted. Typically, the indicates the number of replicates within an experiment but the authors seem to be indicating number of experiments, please clarify.
The number of trial samples is few. How many plots per experiment and how many experiments? Were they conducted in the same year/location.
The results need to improve the text as it is repetitive. The result sections do not need to be written methodically, only the results need to be described. Whether C. rosea can induce resistance against F. graminearum in oat, which should be add the experiment to test, i.e. Ectopic inoculation of C. rosea and F. graminearum to determine whether C. rosea induces resistance to FHB in oat.Minor Revisions:
Line 52: ‘Fungicides sprayed at anthesis can reduce FHB and DON levels in cereals’ Some fungicides can induce DON accumulation, such as carbendazim.
Line 93: C. rosea instead of Clonostachys rosea.
C. rosea and F. graminearum should be italicize throughout the manuscript.
Line 118:C. rosea strain IK726 of FHB changes into C. rosea strain IK726 against FHB.
Line 90: IK726 is the strain of C. rosea, please describe more clearly.The range of variation of the three replicates is somewhat large, and the reproducibility of the trial is questionable (line 125).
Describe The strain LS G2 before using.
Why a significant increase was observed only in trial 3 (line 139).
Trial 1, trial 2, and trial 3 are three repeats, why the difference is great.
All the figures’ letters marked is wrong.
The error bars in the DON control (0 h) is somewhat large, whether the data is reliable. In addition, application of DON concentration is the same, why the difference is large at 0 h between treatment and control.
It is recommended to compare and discuss the changes of pathogenic after biocontrol agent treatment, and the changes of these genes when pathogenic are inoculated alone.
Line 239: ‘neglected’. Please change this word. Pay attention to the accuracy of the words.Is it the amount of all Fusarium or the amount of F. graminearum (Line 125)? In addition, the pictures of the effect of C. rosea against F. graminearum are required.
Line 274-276: ‘The percentage of DON-3-Glc relative to the sum of DON, 3ADON and DON-3 Glc was higher in C. rosea-treated compared to mock treated spikelets, although differences were significant only in one of three trials’ Please discuss it in detail. Line 396: material add s Line 411: mung instead of Mung. Line 412: at instead of AT.Line 414: Please add the whole name when it was referred first time about RCF.

Author Response
In this study, the authors investigated underlying biocontrol effect of biocontrol agent Clonostachys rosea against Fusarium graminearum infection and mycotoxin detoxification in oat by determining Fusarium biomass and DON level. It is meaningful to develop a biocontrol agent which can suppress mycotoxin accumulation of F. graminearum in oat. I think some of authors’ claims are properly supported by data of this manuscript.
<< thank you for your kind comments
Thus, a modification of the language and interpretation of the observation may clarify the true nature of this work for the reader. Major comments: Try to synthetize abstract in a way it is more clear and concise.
<< we have made some adjustments to the abstract
<< Among the co-authors of this manuscript, we happen to have a native speaker with four decades of experience of writing scientific articles and editing books. While revising this manuscript we made sure that this invaluable resource has been used to its fullest – a number of revisions have been made.
In general, improvement of the writing would also help to enhance this work. For example, the description of Clonostachys rosea in the introduction was extremely limited but would have been very helpful to understand the context of the work and why experiments were chosen.
<< we do refer to Clonostachys in the third and last paragraphs of the introduction and refer to a comprehensive review of this organism as a BCA published in 2022. Further specific aspects are brought into the discussion of this paper.
The authors clarified that no reports on the use BCAs to combat the disease in oat production (line 66). The sentence lack of rigor is questionable.
<< a literature search in Web of Science using the search string “(oat or Avena sativa) and Fusarium and (biological control)” gave no hits on biological control of FHB in oat. If we have missed important papers, then please provide the reference(s).
In addition, I have reservations about the interpretation or replication of some of the experiments. It would also help to indicate within each figure legend how many experimental replicates and biological replicates were conducted. Typically, the indicates the number of replicates within an experiment but the authors seem to be indicating number of experiments, please clarify.
<< In the revised version of the manuscript we have clarified information about the replicates. Specifically, that there were 4 replicates in each trial. Each replicate consisted of pooled 15 individually inoculated oat spikelets.
The number of trial samples is few. How many plots per experiment and how many experiments? Were they conducted in the same year/location.
<< these are laboratory/greenhouse trials, not field trials. They were performed at three different time points within 6 months of the same year. We have complemented Materials and Methods section with this information.
The results need to improve the text as it is repetitive. The result sections do not need to be written methodically, only the results need to be described.
<< we have edited the Results section.
Whether C. rosea can induce resistance against F. graminearum in oat, which should be add the experiment to test, i.e. Ectopic inoculation of C. rosea and F. graminearum to determine whether C. rosea induces resistance to FHB in oat.
<< We agree that an experiment showing expression of markers of induced resistance in the system where F. graminearum is inoculated together with C. rosea would strengthen current work . In further work, the results of expression of PR-proteins and WRKYs in system with F.graminearum and C.rosea+F.graminearum will be included. We belive this is beyond the scope of this focussed manuscript.
Minor Revisions:
Line 52: ‘Fungicides sprayed at anthesis can reduce FHB and DON levels in cereals’ Some fungicides can induce DON accumulation, such as carbendazim.
<< yes, indeed
Line 93: C. rosea instead of Clonostachys rosea.
<< corrected
- rosea andF. graminearum should be italicize throughout the manuscript.
<< corrected
Line 118:C. rosea strain IK726 of FHB changes into C. rosea strain IK726 against FHB.
<< corrected
Line 90: IK726 is the strain of C. rosea, please describe more clearly
<< This information is provided in the last paragraph of the Introduction
Describe The strain LS G2 before using.
<< Added more information in Materials and Methods
Why a significant increase was observed only in trial 3 (line 139).
<< Environmental factors would contribute to the DON-detoxifying ability of the plant. We can see that during trial no. 3 (in which values where significant), the Fusarium infection level was the lowest (judging from the amount of Fusarium DNA), so we could speculate that the conditions for infection were not as good as during trials 1 and 2. At the same time, these conditions were favourable for the plant and caused more effective in self-defence, including the expression of DON-detoxifying genes. It could happen that these conditions were also more favourable for better C.rosea endophytic establishment and consequent enhanced induction of DON-glucosylation. As we had the technical capacity to distinguish between DON conjugates, we included DON-3-Glc and 3-ADON into the analysis. This would give us more exact value for the most of DON produced by F. graminearum. The aim of these trials was not to analyse the efficiency of DON-glucosylation, but to see whether we have biocontrol effect or not.
The range of variation of the three replicates is somewhat large, and the reproducibility of the trial is questionable (line 125).
Trial 1, trial 2, and trial 3 are three repeats, why the difference is great.
<< The three greenhouse trials that we conducted could not technically be called „repeats” as they were performed at three different time points (within 6 months of the same year) where plant growing conditions simply could not be 100% identical (temperature, watering, light, humidity). Temperature and weather outside the greenhouse affect conditions in the chamber where oat is grown. The intensity of Fusarium infection is very much dependent on temperature and humidity (and usually positively correlated with both these factors). Besides, hot weather would stress the plant and reduce its ability to withstand the infection (we have seen this in other cereals with F.graminearum). Therefore, the reason why some of the trials showed more infection than others are mostly due to these changing weather conditions. Such differences would be even more pronounced in field trials. What is important here is that, regardless of the infection level and changing environment, we see repeated and robust biological control effect of C.rosea measured by both the amount of Fusarium DNA and the level of mycotoxins produced by the pathogenic fungus. In addition, fungal spore inoculates (both C.rosea and F.graminearum) were grown and prepared anew for each separate trial (which would also add the variation to the system), but this factor did not affect the general biocontrol effect either.
All the figures’ letters marked is wrong
<< we cannot find any errors in these
The error bars in the DON control (0 h) is somewhat large, whether the data is reliable. In addition, application of DON concentration is the same, why the difference is large at 0 h between treatment and control.
<< we agree that the error bars in the DON control (0 h) on Figure 2 are somewhat large. Unfortunately we can not provide any reasonable explanation for that, except that perhaps a technical mistake could have happened during the preparation of the samples, resulting in one of the DON(0h) replicates being confused with water control without DON. Even if DON (0h) values could be better, we believe that differences in the DON(24h) with and without C.rosea treatment are the main valuable results of these experiments.
It is recommended to compare and discuss the changes of pathogenic after biocontrol agent treatment, and the changes of these genes when pathogenic are inoculated alone.
<< We agree that current work lacks the experiment showing expression of markers of induced resistance in the system where F. graminearum is inoculated together with C. rosea. We agree that gene expression experiments F.g. +/- C.r. would be improve the study, but we were constrained with the availability of material, and we chose to focus on other aspects. Should we repeat these experiments, the results of expression of PR-proteins and WRKYs in system with F.graminearum and C.rosea+F.graminearum would be most certainly included.
Line 239: ‘neglected’. Please change this word. Pay attention to the accuracy of the words.
<< we find that this is the most appropriate word to use since our literature searches do not reveal studies on biological control of FHB in oat. This is negligence!
Is it the amount of all Fusarium or the amount of F. graminearum (Line 125)?
<< corrected to F. graminearum
In addition, the pictures of the effect of C. rosea against F. graminearum are required.
<< The symptoms of F.graminearum infection are cryptic, not easily seen on living material, unless seeds are de-hulled. In wheat or barley pictures of the effect of C. rosea against F. graminearum would be illustrative, but as we seen in oat, the symptoms are too variable, and it is not possible to accurately access them visually. As an illustration we have added Figure 2 in supplementary materials, showing an infected oat panicle and kernels both with hulls and de-hulled.
Line 274-276: ‘The percentage of DON-3-Glc relative to the sum of DON, 3ADON and DON-3 Glc was higher in C. rosea-treated compared to mock treated spikelets, although differences were significant only in one of three trials’ Please discuss it in detail.
<< see detailed explanation to the similar question above (Why a significant increase was observed only in trial 3 (line 139).
Line 396: material add s
<< the word “Material” is used in its plural meaning there. This is an indefinite plural.
Line 411: mung instead of Mung.
<< corrected
Line 412: at instead of AT.
<< corrected
Line 414: Please add the whole name when it was referred first time about RCF.
<< we believe this is a standard abbreviation (for relative centrifugal force).
Kind regards,
the authors.
Round 2
Reviewer 1 Report
I believe the current version of the manuscript can be accepted for publication, but some spelling errors still need corrections.
Reviewer 3 Report
First of all, I appreciate that the authors tried their best to improve the manuscript during revision process. However, heere is a issue raised was not properly addressed in the revised manuscript.
Please consult the rules for standard error letters. First, the entire average is arranged from largest to smallest, and then the letter a is marked on the largest average. Instead of the letter a is marked on the minimum average.